# Social isolation but not deprivation involved in employment status after bariatric surgery

Régis Cohen[1]*, Rosa Benvenga[1], Marinos Fysekidis[2], Yasmina Bendacha[1], Jean Marc Catheline[1]

1 Department of Digestive Surgery, Centre Hospitalier de Saint-Denis, Saint-Denis, France, 2 Department of Endocrinology, Hôpital Privé de l'Est Parisien, Aulnay sous Bois, France

☯ These authors contributed equally to this work.
* regis.cohen@ch-stdenis.fr

**Data Availability Statement:** All relevant data are available from the Figshare database (https://figshare.com/search?q=10.6084%2Fm9.figshare.16457595).

## Abstract

An increase in employment rate was observed among individuals who underwent bariatric surgery. This study assessed the relationship between employment rate and weight loss, deprivation, and Bariatric Analysis and Reporting Outcome System (BAROS) scores after bariatric surgery in a deprived area. This retrospective study evaluated the employment rate at a mean period of 2.3±0.1 years after bariatric surgery among 133 individuals. The Evaluation of Deprivation and Inequalities in Health Examination Centers (EPICES score), satisfaction scale, and BAROS (self-esteem, physical activity, social life, work conditions, and sexual activity) questionnaires were used. The mean age of the participants was 45 (range: 19–67) years. Approximately 88% were women. The initial mean body mass index (BMI) was 42.7 kg/m$^2$, and about 88% of the participants underwent sleeve gastrectomy. The mean decrease in BMI was 12 ± 0.5 kg/m$^2$. The mean EPICES score (N<30), BAROS, and satisfaction scale (range: 1–5) scores were 31.9±18, 1.3±1.1, and 4.27±1.19, respectively. After surgery, 19 participants obtained a job. However, three were unemployed. Based on a multivariate analysis, employed and unemployed participants (77 vs 52) before surgery had a lower initial BMI and better BAROS and satisfaction scale scores. After surgery, there was no difference between participants who obtained a new job and those still did not have a job in terms of questionnaire responses. Obtaining a new job was not associated with BMI, sex, or age differences. However, there was a positive correlation between social life score and weight loss. Bariatric surgery increased an individual's chance of finding a job independently of deprivation status. Participants with a pre-operative job had a better perception of satisfaction and BAROS scores. Moreover, social isolation was correlated with unsuccessful weight loss.

## Introduction

The prevalence of obesity is increasing worldwide. The relationship between obesity and complications such as obstructive sleep apnea syndrome, diabetes, hypertension, hyperlipidemia, and nonalcoholic fatty liver disease has been confirmed [1]. Moreover, people with obesity

**Funding:** The author(s) received no specific funding for this work.

**Competing interests:** The authors have declared that no competing interests exist.

have a worse quality of life (QoL), and they present with depression and limitations in activities of daily living [2–5]. Obese patients have a high unemployment rate due to disability from work, absenteeism, and loss of productivity [4, 6–8]. Obesity is associated with direct and indirect costs causing a significant burden on the national health and insurance pension systems [9].

Bariatric surgery was found to be effective in facilitating weight loss, thereby improving related complications and QoL [10, 11]. Whether occupational outcome is an additional benefit from bariatric surgery is not fully recognized. Previous studies evidenced no benefit where as other an improvement rate from 19 to 27% in several systematic [12–14]. Improvement of QoL or psychosocial factors could influence employment rate [15]. However, the relationship between them was not fully evaluated in previous studies.

The current study aimed to assess the impact of bariatric surgery on employment rates during a 2-year follow-up. Moreover, the differences between employed and unemployed participants in terms of QoL (BAROS) and deprivation (EPICES score) were evaluated.

## Materials and methods

### Study design

From our database of patients operated of bariatric surgery we searched eligible participants in our tertiary hospital in the suburbs of Paris, France. This area is characterized by a high rate of immigration and low social economic status of its population. Then, to assess employment status, deprivation, satisfaction, and QoL, we sent a self-questionnaire to 791 patients. All patients underwent bariatric surgery from September 1, 2014, to August 31, 2017, and the questionnaires were collected during follow-up (September 2018).

All patients who underwent different types of bariatric surgery were included. Data on the preoperative baseline characteristics of all participants, including age, gender, weight, and body mass index (BMI), were collected by medical records. Postoperative data on weight, BMI, percentage of total weight loss, and change in BMI were collected by self reporting 2 years after the procedure. Employment status was recorded preoperatively and postoperatively. This information was determined using a questionnaire filled out by the patient before surgery and during follow-up S1 Fig. Retired or permanently disabled patients were excluded from the analysis. We obtained written informed consent from patients during the first medical visit to have data from their medical records used in research. All data were fully anonymized before investigators accessed them. No minors were included during this study. Ethics approval was granted by the local Saint Denis Hospital Ethics Committee and by the national Committee (Commission Nationale informatique et Liberté: ID number: 2220397 v 0).

### Questionnaires

**Assessment of deprivation using EPICES.** The EPICES score Deprivation was assessed using the Evaluation de la Précarité et des Inégalites de santé dans les Centres d'Examens de Santé (Evaluation of Precarity and Inequalities in Health Examination Centers [EPICES]) score computed on the basis of individual conditions of deprivation. The EPICES score Deprivation was assessed using the Evaluation de la Précarité et des Inégalites de santé dans les Centres d'Examens de Santé (Evaluation of Precarity and Inequalities in Health Examination Centers [EPICES]) score computed on the basis of individual conditions of deprivation. Historically, a self-administered questionnaire comprising 42 questions exploring the multiple aspects of deprivation was administered to 7,208 subjects. These socio-economic questions had been selected from bibliographic studies on the determinants of deprivation including traditional socio-economic indicators (education, income, and occupation) as well as questions

related to family structure, housing conditions, employment, social support and benefits, financial difficulties, leisure activities, childhood/adult life events, self-rated health and use of health care. The responses were analyzed by various statistical tools and showed that 11 variables of the 42 questions explained 90% of the variance of the deprivation axis. The EPICES score is derived by summing the estimated regression coefficients corresponding to the responses to the 11 selected questions S2 Fig. The score varied from 0 to 100, from the least deprived to the most deprived situation with a deprivation threshold established at 30 [16]. This EPICES score was validated in 2002 on 200,000 subjects examined in 58 Health Examination Centers French. This score has been shown to be associated with the state of health independently of the French professional or administrative category of deprivation [17].

**Evaluation of QoL using the BAROS questionnaire.** In 1998, the panelists of the National Institutes of Health Consensus Conference panelists developed BAROS. This tool in its original version was used to answer the need to standardize a method for analyzing and reporting QoL after bariatric surgery S3 Fig. The patients reported their QoL using the Likert scale, with scores ranging from −0.50 to 0.50. There were five items for each of the five QoL questions, which were as follows: self-esteem (feel), physical activity, social life, work conditions, and sexual activity [18]. We added the points indicated under each Fig. to calculate the total score and per item.

**Satisfaction questionnaire.** The questionnaire included a 5-item scale, measuring global subjective satisfaction with the surgery. The scale required participants to use absolute ratings. Then, their experience was categorized from 1 to 5, as satisfied (score of 5) or not satisfied (score of 0) S4 Fig.

## Statistical analysis

Quantitative data were presented as mean and standard deviation for normally distributed data. The retired patients were excluded from the analysis (four participants before and five after surgery). The clinical and demographic characteristics of the participants were expressed as mean±standard deviation for continuous variables and percentages for categorical variables. Between groups comparisons were conducted with analysis of variance (ANOVA) and repeated measures ANOVA for quantitative variables and the Fisher's exact test for categorical variables.

In a multivariate analysis including age, gender, BMI, loss of BMI, scores, we compared different groups (A to G) before (A, B) and after surgery (C, D) patients with (A, C) and no job (B, D). Additionally, after surgery we also compare subjects who remained (conservative group E, G) employed (C) or had no (G) to those who obtained a job (E). Conservative job means that they were employed before and after surgery.

## Results

In total, 133 patients responded to our mail. However, four participants who retired from work before surgery were not included in the analysis, and one who retired and another who had missing data after surgery were excluded. The mean age of the participants was 45 (range: 19–67) years, and approximately 88% were women. The pre-operative mean BMI of the participants was 42.7 (range: 33–74) kg/m$^2$, and approximately 88% underwent sleeve gastrectomy. During follow-up, the mean decrease in BMI was 12±0.5 kg/m$^2$. The mean EPICES, BAROS, and satisfaction scale scores were 31.9, 1.3, and 4.27, respectively Table 1. The mean follow-up time was 2.27±0.1 years. In total, 19 participants obtained a job after surgery and three lost their job. Moreover, 77 (60%) of 129 participants had a job before surgery and 92 (72%) of 127 after surgery.

**Table 1. Preoperative baseline characteristics of the population (n = 133).**

| | |
|---|---|
| Average age | 45 (range: 19–67) years |
| **Female/male sex** | 117/16 |
| Weight before surgery (kg) | 115.9±20.5 |
| **Weight after surgery (kg)** | 84±16.7 |
| BMI [a] before surgery | 42.7±6.4 (range: 33–74) kg/m² |
| **BMI [a] after surgery** | 30.88±5.2 (range 33–74) kg/m² |
| Procedure | |
| Sleeve Gastrectomy (SG) | 117 |
| Roux-en-Y gastric bypass | 4 |
| ReSG | 5 |
| SG post lap band | 6 |
| Lap band | 1 |
| EPICES score mean (N < 30.17 = no deprivation) | 31.9±18.1 |
| Score BAROS mean | 1.3±1.1 |

[a] BMI = body mass index

## Before surgery

In total, 77 had a job and 52 did not. The BAROS (p = 0.009) (notably for social, labor dimension) and satisfaction scale (p = 0.02) scores significantly differed between these participants Table 2. Before surgery, the BMI of employed participants significantly inferior (41.7±5 vs 43.6 ±8; p = 0.0001).

## After surgery

After 2 years, 92 had a job (including 19 with a new job), 35 did not have a job, and 3 lost their job. Significant differences were observed between patients who had a job and those who did not have a job in terms of decrease in BMI (p = 0.005) Table 2.

However, there was no significant difference in the different questionnaires between the patients with a new job (n = 19) and those who did not have a job (n = 32).

**Table 2. Preoperative and postoperative questionnaire scores of groups A, B, C, D, E, F, and G.**

| | Before surgery | | After surgery | | | | |
|---|---|---|---|---|---|---|---|
| | A Participants with a job n = 77 | B Participants without a job n = 52 | C Participants with a job n = 92 | D Participants without a job n = 35 | E Participants with a conservative job n = 73 | F Participants with a new job n = 19 | G Participants without a conservative e job n = 32 |
| Decrease BMI[a] ±SD | 11.2±4.2 | 13.1±7.2[AB] | 11.6±4.9 | 13.1±7.3[CD] | 11.3±4.3 | 12.5±6.7[EF] | 13.4±7.6[EG] |
| | NS | <0.0001 | NS | 0.005 | NS | 0.02 | 0.0004 |
| Satisfaction score ±SD | 4.4±0.9 | 3.9±1.2[AB] | 4.4±1.0 | 3.9±1.2 | 4.5±0.9 | 4.1±1.2 | 3.8±1.3[EG] |
| | NS | 0.02 | NS | NS | NS | NS | 0.04 |
| BAROS[b] score ±SD | 1.6±0.9 | 0.9±1.3[AB] | 1.5±1.1 | 0.8±1.2 | 1.6±0.9 | 1.0±1.5[EF] | 0.8±1.2 |
| | NS | 0.009 | NS | NS | NS | 0.02 | NS |
| EPICES[c] score ±SD | 27.8±16.3 | 40.0±17.9 | 29.5±17.8 | 40.1±15.5 | 28.0±16.6 | 36.5±21.4 | 36.5±21.4 |
| | NS | NS | NS | NS | NS | NS | NS |

[a] BMI = body mass index;

[b] EPICES = Evaluation of Deprivation and Inequalities in Health Examination Centers;

[c] BAROS = Bariatric Analysis and Reporting Outcome System. NS means non-significant by comparison between groups before or after surgery

**Table 3. Preoperative and postoperative scores for the different dimensions of BAROS scores in groups A, B, C D, E, F, and G.**

| | Before surgery | | After surgery | | | | |
|---|---|---|---|---|---|---|---|
| | A Participants with a job n = 77 | B Participants without a job n = 52 | C Participants with a job n = 92 | D Participants without a job n = 35 | E Participants with a conservative job n = 73 | F Participants with a new job n = 19 | G Participants without a conservative job n = 32 |
| **Self-esteem** | 0.6±0.4 | 0.4±0.5 | 0.6±0.4 | 0.4±0.5 | 0.6±0.4 | 0.3±0.2 | 0.4±0.5 |
| | NS | NS | NS | NS | NS | NS | NS |
| **Physical** | 0.3±0.2 | 0.1±0.2 | 0.2±0.2 | 0.2±0.2 | 0.3±0.2 | 0.1±0.3 [EF] | 0.2±0.2 |
| | NS | NS | NS | NS | NS | **0.049** | NS |
| **Social life** | 0.3±0.2 | 0.2±0.3 [AB] | 0.3±0.2 | 0.2±0.2 | 0.3±0.2 | 0.2±0.3 [EF] | 0.2±0.3 |
| | NS | **0.0005** | NS | NS | NS | **0.0028** | NS |
| **Labor** | 0.3±0.2 | 0.2±0.3 [AB] | 0.3±0.2 | 0.1±0.3 | 0.3±0.2 | 0.2±0.3 [EF] | 0.1±0.3 |
| | NS | **0.0034** | NS | NS | NS | **0.01** | NS |
| **Sexual** | 0.2±0.3 | 0.0±0.3 | 0.1±0.3 | 0.1±0.3 | 0.2±0.3 | 0±0.4 | 0±0.3 |
| | NS | NS | NS | NS | NS | NS | NS |

BMI = body mass index; EPICES = Evaluation of Deprivation and Inequalities in Health Examination Centers; BAROS = Bariatric Analysis and Reporting Outcome System. NS means non significant by comparison between groups before or after surgery

Only statistically significant differences were indicated between groups.

Patients with a job before surgery (n = 73) and those remaining without (n = 32) had significant differences in decrease of BMI (p = 0.0004) and satisfaction scale score (p = 0.04) Tables 2 and 3. These 73 patients differed also of patients with a new job in decrease of BMI (p = 0.02) and BAROS score (p = 0.02) (notably physical, social life, and labor dimensions: Tables 2 and 3.

The score social life dimension of the BAROS questionnaire, item 3 of S2 Fig., was positively correlated with decrease in BMI (in kg/m$^2$) after surgery (Wilcoxon, <0.0001). The presence of "rich social connections" was associated with better weight loss.

## Discussion

In the current study, after a mean follow-up of 2.3 years after bariatric surgery, there was an improvement in employment status rate by 12% without relation with socio economic status. We identified in participants employed before surgery some significant characteristics: an initial weight inferior, a positive social life/network relationship and better physical capacity. Moreover, a good social score was associated to a better weight loss.

The results of our study are in accordance with those f previous studies for employment rate improvement [19, 20]. It is noteworthy that this study was conducted on a population issued from a region with high unemployment rates [17].

The determinants of this improvement should be investigated. Recently, Kantarovich et al. evaluated changes in mental and physical function based on health-related QoL 2 years after bariatric surgery. Results showed significant improvements in mental and physical function, which account for this change [15]. A recent Danish study evidenced that the post-operative chronic abdominal pain could be an obstacle to employment return [21].

This study aimed to examine whether improvement of employment status is associated with patient characteristics such as age, gender, decrease in BMI, surgical procedure, deprivation status, or some aspects of QoL. Results showed that the employed group was less socially isolated and was more satisfied with bariatric surgery. The importance of social isolation on health status and mortality has been evaluated. A meta-analysis on 308,849 individuals who were followed-up at an average of 7.5 years was conducted. Results showed that individuals

with adequate social relationships have a 50% greater likelihood of survival compared with those with poor or insufficient social relationships. Social relationship is equally or more important than not smoking or other risk factors of mortality (e.g., obesity, physical inactivity). Lack of human contact can also be a predictor of mortality [22]. In our study, we assessed this dimension using question about social life in the BAROS questionnaire. A higher score (less social isolation) was associated with a significantly higher employment rate before and after surgery. Thus, positive social status and relationship could be a factor for employment rate. Moreover, weight loss in our participants was positively linked to the presence of social life of these patients.

The social economic status (EPICES score) did not significantly differ in regards of employment status. Our population has some specificities due to the French culture, health, and social system. This study was performed within an area exposed to precariousness with a higher unemployment rate compared with other areas in the country. Health care and employment rates significantly differed between countries. For example, in Australia, bariatric surgeries are commonly performed in private hospitals. Moreover, they are not fully covered by insurance. Thus, patients must pay an additional fee that cannot be reimbursed. In France, bariatric surgery costs are supported by the healthcare system, and most patient are not charged with any fees. Moreover, the unemployment rates are 5% and 9% respectively in Australia and France. This could explain why despite the presence of low social economic status there was not such effect in our study. The support provided by the French National Health and Social system could buffer the effect of low social economic status [23].

The current study had several limitations. First, a poor response rate of 17%. As noted in previous studies follow up of this population is difficult. It is not excluded that non responders were more likely to be unemployed, have higher deprivation, more comorbidities etc. We did have the means to confirm or not this hypothesis. A retrospective analysis was performed, and the study population was small compared with that of previous studies. Moreover most patients were young women with children, and this could have an impact on employment demands. Ethnic origin, education levels, language, or level of disability were not assessed. It was a short term study and recent publication indicated in another country that increase in the employment rate was not sustained after 7 years of follow-up [24]. From a cultural point of view, some participants might want to stay at home instead of looking for a job, and this point was not evaluated.

## Conclusion

Bariatric surgery had a positive impact on the professional sphere in addition to weight loss and improvement in related diseases and QoL. This return to work was independently of their socio economic status (EPICES score). Participants with a pre-operative job had a better perception of satisfaction and BAROS scores. Moreover, social isolation was correlated with unsuccessful weight loss.

## Supporting information

**S1 Fig. Self-reporting questionnaire for employement.** What was your situation before and after the surgery (circle the correct answer)?.
(TIF)

**S2 Fig. Self-monitoring questionnaire used to evaluate deprivation status (EPICES score).**
(TIF)

**S3 Fig. BAROS questionnaire.** Satisfaction scale. What is your degree of satisfaction with bariatric surgery?. Satisfaction scale. What is your degree of satisfaction with bariatric surgery?. (TIF)

**S4 Fig. Satisfaction scale.** What is your degree of satisfaction with bariatric surgery?. (TIF)

## Author Contributions

**Conceptualization:** Régis Cohen, Marinos Fysekidis, Jean Marc Catheline.

**Formal analysis:** Régis Cohen, Marinos Fysekidis.

**Investigation:** Régis Cohen.

**Methodology:** Régis Cohen, Marinos Fysekidis.

**Resources:** Régis Cohen.

**Supervision:** Régis Cohen, Jean Marc Catheline.

**Validation:** Régis Cohen, Jean Marc Catheline.

**Visualization:** Régis Cohen, Jean Marc Catheline.

**Writing – original draft:** Régis Cohen, Marinos Fysekidis, Jean Marc Catheline.

**Writing – review & editing:** Régis Cohen, Rosa Benvenga, Marinos Fysekidis, Yasmina Bendacha, Jean Marc Catheline.

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
