## [Decision Letter · Decision Letter 0]

13 Apr 2021

PONE-D-20-38829

Social isolation but not deprivation involved in employment status after bariatric surgery.

PLOS ONE

Dear Dr. COHEN,

Thank you for submitting your manuscript to PLOS ONE. After careful consideration, we feel that it has merit but does not fully meet PLOS ONE’s publication criteria as it currently stands. Therefore, we invite you to submit a revised version of the manuscript that addresses the points raised during the review process.

An expert in the field handled your manuscript, and we are appreciative of their time and for carefully reading your manuscript. Although interest was found in your study, several major concerns arose that require your attention. All sections of this manuscript, including the data presentation, require major revisions. Please address ALL of the reviewer's comments in your revised manuscript and rebuttal letter.

Submit your revised manuscript by May 22 2021 11:59PM. If you will need more time than this to complete your revisions, please reply to this message or contact the journal office at plosone@plos.org. Please include the following items when submitting your revised manuscript:

We look forward to receiving your revised manuscript.

Kind regards,

Frank T. Spradley

Academic Editor

PLOS ONE

3. We note that you state in your ethics statement in trhe manuscript text and online submission form that "We obtained written informed consent from patients during the first medical visit to have data from their medical records used in research" and "All data were fully anonymized before investigators accessed them." Please clarify how written informed consent was obtained from patients while still keeping the data anonymized before you accessed them.

4. Please provide all p-values in Tables 2 and 3, even if they are not significant.

5. We note that you state in your discussion "from an ethnic (African or North African origins) or cultural point of view, the participants might want to stay at home instead of looking for a job, and this point was not evaluated." As it is possible that a reader may construe this as reinforcing harmful stereotypes about black people, we ask that this sentence be revised. Please either:

(1) provide several citations that support the claim that African people might want to stay home instead of looking for a job,

(2) remove this sentence, or

(3) revise the sentence to state "from a cultural point of view, some participants might want to stay at home instead of looking for a job, and this point was not evaluated.

6. We note that you have indicated that data from this study are available upon request. PLOS only allows data to be available upon request if there are legal or ethical restrictions on sharing data publicly. For information on unacceptable data access restrictions, please see http://journals.plos.org/plosone/s/data-availability#loc-unacceptable-data-access-restrictions.

Reviewers' comments:

Reviewer's Responses to Questions

**Comments to the Author**

1. Is the manuscript technically sound, and do the data support the conclusions?

Reviewer #1: Partly

2. Has the statistical analysis been performed appropriately and rigorously? 

Reviewer #1: No

3. Have the authors made all data underlying the findings in their manuscript fully available?

Reviewer #1: No

4. Is the manuscript presented in an intelligible fashion and written in standard English?

Reviewer #1: Yes

5. Review Comments to the Author

Reviewer #1: This study aimed to assess the impact of bariatric surgery on employment rates during a 2-year follow-up and to look at the differences between employed and unemployed participants in terms of quality of life and deprivation scores. To address the research question the authors invited 791 patients who had undergone bariatric surgery to participate, only 133 patients responded to the questionnaire, and a further 6 were excluded, leaving 127 participants (17% response rate). Between-group comparisons were conducted with ANOVA and repeated measures ANOVA and Fisher’s exact test. There was an increase in the number of employed participants 2 years after surgery. The deprivation score did not differ between employed and unemployed participants before or after surgery. Quality of life was lower for participants without a job (than those with a job) before surgery. After surgery QoL was lower for participants with a new job compared to those who retained their job.

Overall, the authors need to provide additional justification of the need for the study and what the study adds to the existing literature, particularly given the small sample size (and low response rate). In addition, the paper requires significant editing and proof-reading.

Major issues:

Abstract:

Lines 16-17: Background: This section appears to state the results of the analysis, rather than providing a rationale for the study and its overall aims. It is also unclear how the range of the increase in the employment rate has been calculated (i.e. 17-29%). These are results are not included in the body of the manuscript.

The results presented in the Abstract are very confusing and the conclusions are not supported by the results presented.

Introduction:

Lines 52-53: References should be provided for the sentence stating that there are inconsistent results from studies looking at the occupational outcomes of bariatric surgery. This statement would benefit from an extra sentence or two summarising what these inconsistencies were. The statement then seems to be contradicted by the following sentence stating that two systematic reviews have concluded that bariatric surgery results in better employment outcomes.

Lines 55-56: References should also be provided for these statements. It is unclear whether the authors are talking about past studies that have looked at the associations between QoL and deprivation and employment status in people who have had bariatric surgery or in wider population groups. These statements appear to be the main rationale for the study; however, how this study will add to the existing literature has not been well argued.

Methods:

Study design

Line 62: Further details on the study location and setting should be provided e.g. the hospital in which the study took place, the location of the hospital in France and the general characteristics of the location/region where the hospital is located (e.g. employment rate, extent of social disadvantage) and how these characteristics might differ from France as a whole.

Line 62: Could the authors please clarify what criteria defined an “eligible participant”. How many participants were excluded because they did not meet these criteria?

Line 67: Could the authors please clarify whether the preoperative characteristics of age, gender, weight, and BMI were collected by self-report or abstracted from medical records? Were the postoperative characteristics (particularly weight and BMI) collected in the same way, or were these collected by self-report? If these characteristics were measured in different ways before and after surgery this should be mentioned as a limitation in the Discussion.

Line 70: Could the authors please provide (either in the text of the manuscript or in the supplementary material) the specific employment questions asked in both the pre-operative and post-operative surveys.

Evaluation of QoL using the BAROS questionnaire

Line 89: Could the authors please clarify how they are using the BAROS questionnaire, and also why they appear to have used the original version and not the updated version (Oria HE, Moorehead MK. Updated Bariatric Analysis and Reporting Outcome System (BAROS). Surg Obes Relat Dis. 2009 Jan-Feb;5(1):60-6. doi: 10.1016/j.soard.2008.10.004. Epub 2008 Nov 1. PMID: 19161935.)?

Lines 90-94: The methods indicate that the authors have only used the quality of life component of the BAROS and not the weight-loss and change in co-morbidities components. If this is the case, can the authors provide information on why they have done this and whether it is a validated approach to only use parts of the BAROS; and how the scoring is then able to be interpreted?

Satisfaction Scale

Lines 96-98: Could the authors please clarify how patients who selected the middle points of the Satisfaction Scale (i.e. 2,3,4) were categorised in the analysis? The text gives the impression that only participants that selected responses of 0 and 5 were categorised.

Statistical analysis

Lines 100-102: The description of the statistical analysis seems to indicate that different numbers of participants have been included in the pre- and postoperative groups (four retired patients excluded from the preoperative group and five from the postoperative group). There should be the same number of patients in both groups i.e. participants who were retired either pre- or postoperatively should have been excluded from all analyses.

Results

The authors should make sure in the results section that they answer the specific aims/research questions of the study.

Line 112: A response rate of 17% is extremely low (133/791 patients). This should be addressed more fully in the limitations section of the Discussion as it is possible that people who did not respond to the questionnaire were more likely to be unemployed, have higher deprivation, more comorbidities etc.

Lines 152-154: More detailed information on the correlations between decrease in BMI and the social dimension of the BAROS would be useful as the authors discuss this at length in the Discussion and this is one of the key conclusions of the paper (even though it is not one of the stated aims of the study). Would these differences be considered clinically significant?

Discussion

Lines 156-160: The summary of the results at the beginning of the Discussion do not adequately reflect the aims of the paper.

Lines 161-168: The discussion about previous studies does not provide sufficient information on the similarities/differences in results with other studies or the reasons for why differences may be present.

Lines 195-200: As noted above, the low response rate should be highlighted as a limitation of the study. In addition, another limitation that the authors allude to, but do not clearly state, appears to be that it is unknown whether participants were actually seeking employment during the follow-up period e.g. they note that many of the participants were women with young children. It is a major limitation if participants who were not actively looking for work were unable to be excluded from the study.

Conclusion

Line 202: The concluding statement notes that bariatric surgery had a positive impact on improvement in related diseases. I am not clear that the study measured this association.

Lines 203-204: Could the authors clarify what is meant by this sentence. The study did not appear to have information on whether unemployed participants were provided with opportunities to return to work. The study only found that there were no statistically significant differences in the deprivation score between employed and unemployed participants.

Minor issues:

Introduction:

Lines 43-45: references should be provided for the claims made about associations between obesity and the diseases listed.

Assessment of deprivation using EPICES

Lines 80-82: The wording of this paragraph is confusing. It reads as though the authors reduced the set of EPICES questions; however, further reading shows that this is not the case. It would be useful to provide further background on the EPICES measure of deprivation for the benefit of international readers who may not be familiar with it.

Results

Lines 115-117: the authors should be specific when describing the results as to whether they are referring to pre- or post-operative characteristics e.g. the mean BMI of 42.7.

Table 1

I think the third line of the table labelled “weight before surgery (kg)” is probably supposed to read “weight after surgery (kg)”.

It would be preferable to spell out the in full in the body of the table rather than in a footnote (there is plenty of space to do so).

Could the authors please clarify what (N<30.17) against Score EPICES score is meant to indicate in the context of the Table?

Table 2

The term “conservative job” is not a common English term. Do the authors mean Participants who were in the same job pre- and post-operatively?

6. PLOS authors have the option to publish the peer review history of their article (what does this mean?). If published, this will include your full peer review and any attached files.

Reviewer #1: No

---

## [Author Response · Author response to Decision Letter 0]

9 Jun 2021

Frank T. Spradley

Academic Editor

PLOS ONE

Revision required 

PONE-D-20-38829

Social isolation but not deprivation involved in employment status after bariatric surgery.

PLOS ONE

You will find the rebuttal letter that responds to each point raised by the academic editor and reviewer(s). You should upload this letter as a separate file labeled 'Response to Reviewers'.

And attached A marked-up copy of our manuscript and the unmarked version of our revised paper 

Kind regards,

I think we did the job

We worked with enago to be sure of the language. You will find the certificate 

Régis Cohen and Marnos Fysekidis 

3. We note that you state in your ethics statement in trhe manuscript text and online submission form that "We obtained written informed consent from patients during the first medical visit to have data from their medical records used in research" and "All data were fully anonymized before investigators accessed them." Please clarify how written informed consent was obtained from patients while still keeping the data anonymized before you accessed them.

Yes we confirm that the XLS table used is completely anonymized and received the agreement of our local and national ethical committee

4. Please provide all p-values in Tables 2 and 3, even if they are not significant.

We indicate by NS the non-significant result in Table 2 and 3 

We changed 5. We note that you state in your discussion "from an ethnic (African or North African origins) or cultural point of view, the participants might want to stay at home instead of looking for a job, and this point was not evaluated." As it is possible that a reader may construe this as reinforcing harmful stereotypes about black people, we ask that this sentence be revised. Please either:

(1) provide several citations that support the claim that African people might want to stay home instead of looking for a job,

(2) remove this sentence, or

(3) revise the sentence to state "from a cultural point of view, some participants might want to stay at home instead of looking for a job, and this point was not evaluated.

We thank you for the suggested sentence that we used in the text : “from a cultural point of view, some participants might want to stay at home instead of looking for a job, and this point was not evaluated”.

6. We note that you have indicated that data from this study are available upon request. PLOS only allows data to be available upon request if there are legal or ethical restrictions on sharing data publicly. For information on unacceptable data access restrictions, please see http://journals.plos.org/plosone/s/data-availability#loc-unacceptable-data-access-restrictions.

We read it and we think we are in accordance with legal or ethical restrictions

No 

We did it 

7. Please include captions for your Supporting Information files at the end of your manuscript, and update any in-text citations to match accordingly. Please see our Supporting Information guidelines for more information: http://journals.plos.org/plosone/s/supporting-information<about:blank>.

Done

1. Is the manuscript technically sound, and do the data support the conclusions?

Reviewer #1: Partly 

We changed the conclusion messages at the end of discussion

2. Has the statistical analysis been performed appropriately and rigorously?

Reviewer #1: No 

We added some information in presentation of table and in the text to explain the methods of statistical analysis

3. Have the authors made all data underlying the findings in their manuscript fully available?

The PLOS Data policy<http://www.plosone.org/static/policies.action#sharing> requires authors to make all data underlying the findings described in their manuscript fully available without restriction, with rare exception (please refer to the Data Availability Statement in the manuscript PDF file). The data should be provided as part of the manuscript or its supporting information, or deposited to a public repository. For example, in addition to summary statistics, the data points behind means, medians and variance measures should be available. If there are restrictions on publicly sharing data—e.g. participant privacy or use of data from a third party—those must be specified.

Reviewer #1: No 

We read and modified the text 

4. Is the manuscript presented in an intelligible fashion and written in standard English?

Reviewer #1: Yes 

OK 

5. Review Comments to the Author

Reviewer #1: This study aimed to assess the impact of bariatric surgery on employment rates during a 2-year follow-up and to look at the differences between employed and unemployed participants in terms of quality of life and deprivation scores. To address the research question the authors invited 791 patients who had undergone bariatric surgery to participate, only 133 patients responded to the questionnaire, and a further 6 were excluded, leaving 127 participants (17% response rate). Between-group comparisons were conducted with ANOVA and repeated measures ANOVA and Fisher’s exact test. There was an increase in the number of employed participants 2 years after surgery. The deprivation score did not differ between employed and unemployed participants before or after surgery. Quality of life was lower for participants without a job (than those with a job) before surgery. After surgery QoL was lower for participants with a new job compared to those who retained their job.

Overall, the authors need to provide additional justification of the need for the study and what the study adds to the existing literature, particularly given the small sample size (and low response rate). In addition, the paper requires significant editing and proof-reading.

Major issues:

Abstract:

Lines 16-17: Background: This section appears to state the results of the analysis, rather than providing a rationale for the study and its overall aims. It is also unclear how the range of the increase in the employment rate has been calculated (i.e. 17-29%). These are results are not included in the body of the manuscript.

This data is extracted from Charras, L., Savall, F., Descazaux, T., Soulat, J. M., Ritz, P., & Herin, F. (2017). Comment on: systematic review and meta-analysis of occupational outcomes after bariatric surgery. Obesity surgery, 27(3), 811. We agree with reveiwer that is confusing and not a pertinent way to indicate this evidence in an abstract . We delete the information 

The results presented in the Abstract are very confusing and the conclusions are not supported by the results presented.

Introduction:

Lines 52-53: References should be provided for the sentence stating that there are inconsistent results from studies looking at the occupational outcomes of bariatric surgery. This statement would benefit from an extra sentence or two summarising what these inconsistencies were. The statement then seems to be contradicted by the following sentence stating that two systematic reviews have concluded that bariatric surgery results in better employment outcomes.

We change the sentences

Lines 55-56: References should also be provided for these statements. It is unclear whether the authors are talking about past studies that have looked at the associations between QoL and deprivation and employment status in people who have had bariatric surgery or in wider population groups. These statements appear to be the main rationale for the study; however, how this study will add to the existing literature has not been well argued.

We change the sentences and add the reference

Methods:

Study design

Line 62: Further details on the study location and setting should be provided e.g. the hospital in which the study took place, the location of the hospital in France and the general characteristics of the location/region where the hospital is located (e.g. employment rate, extent of social disadvantage) and how these characteristics might differ from France as a whole.

Line 62: Could the authors please clarify what criteria defined an “eligible participant”. How many participants were excluded because they did not meet these criteria?

We change for: “From our database of patients operated of bariatric surgery we searched eligible participants in our tertiary hospital in the suburbs of Paris, France. This area is characterized by a high rate of immigration and low social economic status of its population.”

Line 67: Could the authors please clarify whether the preoperative characteristics of age, gender, weight, and BMI were collected by self-report or abstracted from medical records? Were the postoperative characteristics (particularly weight and BMI) collected in the same way, or were these collected by self-report? If these characteristics were measured in different ways before and after surgery this should be mentioned as a limitation in the Discussion.

Yes they were medically recorded before surgery and self-reported after surgery (we changed the sentences) 

Line 70: Could the authors please provide (either in the text of the manuscript or in the supplementary material) the specific employment questions asked in both the pre-operative and post-operative surveys.

Yes we added a supporting material with the form sent to patients S1 Fig.

S1 Fig. Self-reporting questionnaire for employement. 

What was your situation before and after the surgery (circle the correct answer)?

Before surgery 

retirement

employment

 Without work for more than 6 months 

Unemployment

 Work stoppage due to illness

After surgery 

retirement

employment

 Without work for more than 6 months 

Unemployment

 Work stoppage due to illness

Evaluation of QoL using the BAROS questionnaire

Line 89: Could the authors please clarify how they are using the BAROS questionnaire, and also why they appear to have used the original version and not the updated version (Oria HE, Moorehead MK. Updated Bariatric Analysis and Reporting Outcome System (BAROS). Surg Obes Relat Dis. 2009 Jan-Feb;5(1):60-6. doi: 10.1016/j.soard.2008.10.004. Epub 2008 Nov 1. PMID: 19161935.)?

Lines 90-94: The methods indicate that the authors have only used the quality of life component of the BAROS and not the weight-loss and change in co-morbidities components. If this is the case, can the authors provide information on why they have done this and whether it is a validated approach to only use parts of the BAROS; and how the scoring is then able to be interpreted?

We have been using the original version of this score since 2003 out of habit. We had not noticed its most recent modification. We just added the points indicated under each figure to calculate the total score and per item (as indicated in supplementary material). We indicated in the text that we used the original version and the method of calculation

Satisfaction Scale

Lines 96-98: Could the authors please clarify how patients who selected the middle points of the Satisfaction Scale (i.e. 2,3,4) were categorised in the analysis? The text gives the impression that only participants that selected responses of 0 and 5 were categorised.

Thank you for this comment we modified the sentence 

Statistical analysis

Lines 100-102: The description of the statistical analysis seems to indicate that different numbers of participants have been included in the pre- and postoperative groups (four retired patients excluded from the preoperative group and five from the postoperative group). There should be the same number of patients in both groups i.e. participants who were retired either pre- or postoperatively should have been excluded from all analyses.

As indicated they were excluded of the analysis in pre or post operative analysis Lines 100-102 modified manuscript 

Results

The authors should make sure in the results section that they answer the specific aims/research questions of the study.

Line 112: A response rate of 17% is extremely low (133/791 patients). This should be addressed more fully in the limitations section of the Discussion as it is possible that people who did not respond to the questionnaire were more likely to be unemployed, have higher deprivation, more comorbidities etc.

We add sentences in the limitation paragraph of discussion: “First, a poor response rate of 17 %. As noted in previous studies follow up of this population is difficult. It is not excluded that non responders were more likely to be unemployed, have higher deprivation, more comorbidities etc. We did have the means to confirm or not this hypothesis.”

Lines 152-154: More detailed information on the correlations between decrease in BMI and the social dimension of the BAROS would be useful as the authors discuss this at length in the Discussion and this is one of the key conclusions of the paper (even though it is not one of the stated aims of the study). Would these differences be considered clinically significant?

We hope to have clarified this point by new sentences : The score social life dimension of the BAROS questionnaire (item 3 of Supplementary Figure 2 BAROS questionnaire ) was positively correlated with decrease in BMI (in kg/m2) after surgery (Wilcoxon, <0.0001). The presence of “rich social connections” was associated with better weight loss. 

Discussion

Lines 156-160: The summary of the results at the beginning of the Discussion do not adequately reflect the aims of the paper.

We change the first paragraph for : “In the current study, after a mean follow-up of 2.3 years after bariatric surgery, there was an improvement in employment status rate by 12% without relation with socio economic status. We identified in participants employed before surgery some significant characteristics: an initial weight inferior, a positive social life/network relationship and better physical capacity. Moreover, a good social score was associated to a better weight loss.”

Lines 161-168: The discussion about previous studies does not provide sufficient information on the similarities/differences in results with other studies or the reasons for why differences may be present.

We did some changes to improve the message

Lines 195-200: As noted above, the low response rate should be highlighted as a limitation of the study. In addition, another limitation that the authors allude to, but do not clearly state, appears to be that it is unknown whether participants were actually seeking employment during the follow-up period e.g. they note that many of the participants were women with young children. It is a major limitation if participants who were not actively looking for work were unable to be excluded from the study.

We did changes 

Conclusion

Line 202: The concluding statement notes that bariatric surgery had a positive impact on improvement in related diseases. I am not clear that the study measured this association.

We changed the conclusion 

Lines 203-204: Could the authors clarify what is meant by this sentence. The study did not appear to have information on whether unemployed participants were provided with opportunities to return to work. The study only found that there were no statistically significant differences in the deprivation score between employed and unemployed participants.

We clarified this point by changindg the conclusion : “Bariatric surgery had a positive impact on the professional sphere in addition to weight loss and improvement in related diseases and QoL. This return to work was independently of their socio economic status (EPICES score).. Participants with a pre-operative job had a better perception of satisfaction and BAROS scores. Moreover, social isolation was correlated with unsuccessful weight loss. »

Minor issues:

Introduction:

Lines 43-45: references should be provided for the claims made about associations between obesity and the diseases listed.

Yes we added a reference 

Assessment of deprivation using EPICES

Lines 80-82: The wording of this paragraph is confusing. It reads as though the authors reduced the set of EPICES questions; however, further reading shows that this is not the case. It would be useful to provide further background on the EPICES measure of deprivation for the benefit of international readers who may not be familiar with it.

We detailed the conception of this questionnaire “The EPICES score Deprivation was assessed using the Evaluation de la Précarite´ et des Inégalites de sante´ dans les Centres d’Examens de Sante´ (Evaluation of Precarity and Inequalities in Health Examination Centers [EPICES]) score computed on the basis of individual conditions of deprivation. A first questionnaire exploring the multiple aspects of deprivation was administered to 7,208 subjects aged 16–59 years examined at 18 French Health Examinations Centers in 1999. The 42 socioeconomic questions selected from a bibliographic study on determinants of deprivation included traditional socioeconomic indicators (education, income, and occupation) and also questions related to family structure, housing conditions, employment, social benefits, financial difficulties, leisure activities, social support, childhood/adult life events, self-perceived health, and health care utilization. Data from the questionnaire were first analyzed by correspondence analysis [13]. To reduce the number of variables, multiple regression analysis was used to select a subset of indicator variables, giving a satisfactory explanation of the individual coordinate on the principal axis associated with deprivation. The regression analysis showed that 11 variables of the 42 questions explained 90% of the variance of the deprivation axis. The EPICES score is derived by summing the estimated regression coefficients corresponding to the responses to the 11 selected questions (Supplementary Figure 1). The score varied from 0 to 100, from the least deprived to the most deprived situation. The EPICES score was validated in 2002 in 200,000 subjects examined at 58 French Health Examination Centers. The EPICES score was found to be associated with health status independently of the occupational category and the French administrative definition of deprivation [14].

Results

Lines 115-117: the authors should be specific when describing the results as to whether they are referring to pre- or post-operative characteristics e.g. the mean BMI of 42.7.

We did it 

Table 1

I think the third line of the table labelled “weight before surgery (kg)” is probably supposed to read “weight after surgery (kg)”.

Sorry for error corrected 

It would be preferable to spell out the in full in the body of the table rather than in a footnote (there is plenty of space to do so).

We didi it 

Could the authors please clarify what (N<30.17) against Score EPICES score is meant to indicate in the context of the Table?

We did it in the table 

Table 2

The term “conservative job” is not a common English term. Do the authors mean Participants who were in the same job pre- and post-operatively?

You are right we do not have this information conservative job means that they are employed before and after surgery. We added this explanation in the text

6. PLOS authors have the option to publish the peer review history of their article (what does this mean?<https://journals.plos.org/plosone/s/editorial-and-peer-review-process#loc-peer-review-history>). If published, this will include your full peer review and any attached files.

We accept this option “Yes”

Do you want your identity to be public for this peer review? For information about this choice, including consent withdrawal, please see our Privacy Policy<https://www.plos.org/privacy-policy>.

Reviewer #1: No

While revising your submission, please upload your figure files to the Preflight Analysis and Conversion Engine (PACE) digital diagnostic tool, https://pacev2.apexcovantage.com/. PACE helps ensure that figures meet PLOS requirements. To use PACE, you must first register as a user. Registration is free. Then, login and navigate to the UPLOAD tab, where you will find detailed instructions on how to use the tool. If you encounter any issues or have any questions when using PACE, please email PLOS at figures@plos.org<mailto:figures@plos.org>. Please note that Supporting Information files do not need this step.

We hope to be in agreement with this. We did the analysis of our manuscript By PACE 

In compliance with data protection regulations, you may request that we remove your personal registration details at any time. (Remove my information/details)<https://www.editorialmanager.com/pone/login.asp?a=r>. Please contact the publication office if you have any questions.

---

## [Decision Letter · Decision Letter 1]

5 Jul 2021

PONE-D-20-38829R1

Social isolation but not deprivation involved in employment status after bariatric surgery.

PLOS ONE

Dear Dr. COHEN,

Thank you for submitting your manuscript to PLOS ONE. After careful consideration, we feel that it has merit but does not fully meet PLOS ONE’s publication criteria as it currently stands. Therefore, we invite you to submit a revised version of the manuscript that addresses the points raised during the review process.

We look forward to receiving your revised manuscript.

Kind regards,

Frank T. Spradley

Academic Editor

PLOS ONE

Journal Requirements:

Reviewers' comments:

Reviewer's Responses to Questions

**Comments to the Author**

1. If the authors have adequately addressed your comments raised in a previous round of review and you feel that this manuscript is now acceptable for publication, you may indicate that here to bypass the “Comments to the Author” section, enter your conflict of interest statement in the “Confidential to Editor” section, and submit your "Accept" recommendation.

Reviewer #1: (No Response)

2. Is the manuscript technically sound, and do the data support the conclusions?

Reviewer #1: Yes

3. Has the statistical analysis been performed appropriately and rigorously? 

Reviewer #1: Yes

4. Have the authors made all data underlying the findings in their manuscript fully available?

Reviewer #1: Yes

5. Is the manuscript presented in an intelligible fashion and written in standard English?

Reviewer #1: Yes

6. Review Comments to the Author

Reviewer #1: The authors have adequately addressed the reviewer comments; however, in addressing the request for additional information on the EPICES measure, the authors have copy and pasted paragraphs from Bihan et. al.: Association among individual deprivation, glycemic control, and diabetes complications: the EPICES score. Diabetes Care. 2005 Nov;28(11):2680-5. doi: 10.2337/diacare.28.11.2680. PMID: 16249539.

Although the authors have cited this reference, a direct copy and paste is not appropriate.

7. PLOS authors have the option to publish the peer review history of their article (what does this mean?). If published, this will include your full peer review and any attached files.

Reviewer #1: No

---

## [Author Response · Author response to Decision Letter 1]

17 Aug 2021

Dear Editor 

First of all, we thank you for the quality of the proofreading work done by your teams.

We are sorry for the shortcomings during the first proofreading and we hope to have met your expectations.

Pease find the answers to your request 

ONE-D-20-38829R1

Social isolation but not deprivation involved in employment status after bariatric surgery.

PLOS ONE

Dear Dr. COHEN,

Thank you for submitting your manuscript to PLOS ONE. After careful consideration, we feel that it has merit but does not fully meet PLOS ONE’s publication criteria as it currently stands. Therefore, we invite you to submit a revised version of the manuscript that addresses the points raised during the review process.

We did it 

We do not have laboratory protocol?? 

We look forward to receiving your revised manuscript.

Kind regards,

Frank T. Spradley

Academic Editor

PLOS ONE

Journal Requirements:

Reviewers' comments:

Reviewer's Responses to Questions

Comments to the Author

1. If the authors have adequately addressed your comments raised in a previous round of review and you feel that this manuscript is now acceptable for publication, you may indicate that here to bypass the “Comments to the Author” section, enter your conflict of interest statement in the “Confidential to Editor” section, and submit your "Accept" recommendation.

Reviewer #1: (No Response)

2. Is the manuscript technically sound, and do the data support the conclusions?

 Reviewer #1: Yes

We thank you 

3. Has the statistical analysis been performed appropriately and rigorously?

 Reviewer #1: Yes

We thank you 

4. Have the authors made all data underlying the findings in their manuscript fully available?

Reviewer #1: Yes

We thank you 

 5. Is the manuscript presented in an intelligible fashion and written in standard English?

 Reviewer #1: Yes

We thank you 

6. Review Comments to the Author

Reviewer #1: The authors have adequately addressed the reviewer comments; however, in addressing the request for additional information on the EPICES measure, the authors have copy and pasted paragraphs from Bihan et. al.: Association among individual deprivation, glycemic control, and diabetes complications: the EPICES score. Diabetes Care. 2005 Nov;28(11):2680-5. doi: 10.2337/diacare.28.11.2680. PMID: 16249539.

Although the authors have cited this reference, a direct copy and paste is not appropriate.

In this paragraph we have attempted to respond to the comments of the reviewers. For that we used a paragraph published by us and accepted describing very precisely our questionnaire. We understand the new remarks and we have changed the text while keeping the main part of the description of the EPICES questionnaire

7. PLOS authors have the option to publish the peer review history of their article (what does this mean?). If published, this will include your full peer review and any attached files.

Do you want your identity to be public for this peer review? For information about this choice, including consent withdrawal, please see our Privacy Policy.

We accept to publish the peer review history of our manuscript. We have nothing to hide, no conflict of interest, our protocol is ethical (approved) and the data is anonymized.

Reviewer #1: No

We did it 

Kind regards,

Régis Cohen 

MD, PhD

---

## [Editor Report · Decision Letter 2]

20 Aug 2021

Social isolation but not deprivation involved in employment status after bariatric surgery.

PONE-D-20-38829R2

Dear Dr. COHEN,

We’re pleased to inform you that your manuscript has been judged scientifically suitable for publication and will be formally accepted for publication once it meets all outstanding technical requirements.

Kind regards,

Frank T. Spradley

Academic Editor

PLOS ONE

---

## [Editor Report · Acceptance letter]

31 Aug 2021

PONE-D-20-38829R2 

Social isolation but not deprivation involved in employment status after bariatric surgery. 

Dear Dr. Cohen:

I'm pleased to inform you that your manuscript has been deemed suitable for publication in PLOS ONE. Congratulations! Your manuscript is now with our production department. 

Kind regards, 

on behalf of

Dr. Frank T. Spradley 

Academic Editor

PLOS ONE